# National Audit, Media Attention, and Efficiency of Local Fiscal Expenditure: A Spatial Econometric Analysis Based on Provincial Panel Data in China

**Dingzu Zhang** , **Xingjie Shen and Cong Peng** *

School of Economics and Management, Changsha University of Science and Technology, Changsha 410004, China
*   Correspondence: cljgpc@163.com

**Abstract:** Improving the efficiency of local fiscal expenditure is an important way to swiftly mitigate local fiscal risks according to the current economic situation. Based on the provincial panel data of 30 provinces in mainland China (except Tibet Autonomous Region) from 2007 to 2018, a Tobit spatial error model was constructed to test the impact of national auditing on local fiscal expenditure efficiency and to investigate the intermediary role of media attention. Findings show that the disclosing, resisting, and preventing functions of the national audit significantly improve local fiscal expenditure efficiency. Media attention does play an intermediary role, indicating that the information transmission function of the national audit has governance effects. Coupling of the national audit and media attention also positively affects local fiscal expenditure efficiency. This research expands the mechanism of national audits, as they affect the efficiency of local fiscal expenditure; it also provides new empirical evidence for improving such efficiency while mitigating fiscal risks at the local level.

**Keywords:** national audit; local fiscal expenditure efficiency; media attention; disclosure of national audit results





## 1. Introduction

At present, China's fiscal situation is, arguably, grim. Local fiscal pressure is increasing, and local fiscal sustainability is threatened by the domestic economic slowdown and the continuous upside-down growth of fiscal revenue and expenditure as well as by external shocks such as the continued spread of the global epidemic, economic downturn, and increased risks [1]. Local fiscal risks can essentially be attributed to inefficient fiscal expenditure, and the core of local fiscal risks is the efficiency of local fiscal expenditure [2]. The efficiency of local fiscal expenditure must be improved to alleviate local fiscal pressure, defuse risks, and promote the sustainable development of local fiscal infrastructures [3].

National auditing, a risk-control measure enacted via national governance, allows for independent supervision according to law. Auditing the allocation, implementation, use, and management of the government budget improves the performance of fiscal funds as well as the budget execution ability at the unit level [4], thus improving the efficiency of local fiscal expenditure. A national audit not only positively affects local government debt risk [5,6], fiscal security [7,8], budget transparency [9], fiscal revenue, expenditure violations [10–12], budget management [13], budget violations [14], and fiscal sustainability [15] but also has a positive governance effect on local fiscal expenditure efficiency [16,17].

The disclosure (i.e., information transmission) of national audit results is conducive to organic coordination across various modes of supervision and further amplifies the governance effect of auditing [18]. For example, the first announcement of audit results released by the National Audit Office in 2003 aroused media attention and set off an "audit storm". The research holds that the national audit, with the result and work reports as the information carrier, transmits to the society the running track and results of the public

power of the users of fiscal funds and resources. This provides a basis for the government to strengthen budget management, for the National People's Congress (NPC) to strengthen budget supervision, and for the media to participate in fiscal supervision [19]. The media's attention on and dissemination of audit results stimulate more public participation in fiscal supervision, which has led to more effective governance over many fiscal expenditure violations [20] and enhanced the effectiveness of supervision [21]. This is, essentially, the governance effect of national audit information transmission function. Despite valuable research on this subject, a lack of in-depth analysis has left significant knowledge gaps.

In the present study, we first examined the direct effects of national auditing on the efficiency of local fiscal expenditure according to disclosing, resisting, and preventing functions. We then examined the mediating effect of media attention on the national audit in improving local fiscal expenditure efficiency. The results show that the disclosing, resisting, and preventing functions of the national audit significantly improve local fiscal expenditure efficiency; media attention plays a part of mediating effect; and the coupling level of national audit and media attention is positively correlated with local fiscal expenditure efficiency.

## 2. Literature Review

### 2.1. Governance Effect of National Auditing on Local Fiscal Expenditure Efficiency

There may be illegal use, loss, and waste in the process of local governmental fiscal expenditure resulting in efficiency loss. National auditing is an important system under which revenues and expenditures of local governments are supervised according to law. Various auditing mechanisms reveal, resist, or prevent elements of the distribution of the local government budgets. Audits also may supervise the implementation, use, and management of funds; they may disclose irregularities across auditees and reduce corruption caused by factors such as waste. High-quality audit recommendations can standardize the use of fiscal capital, improving the capacity for budget implementation and performance [4], the use of public funds to improve the fiscal funding levels given output, or saving fiscal funds under the given level of output so as to promote efficiency of fiscal expenditure. An effective national audit is considered in this paper to be a direct improvement to local fiscal expenditure efficiency.

The World Bank and INTOSAI hold that national auditing improves fiscal expenditure efficiency by holding public fiscal resources accountable, reducing fund abuse, and strengthening system construction. Previous scholars have found that a national audit significantly improves the degree of fiscal security in local governments [22] and plays a role in correcting budget deviations [23]. These results intensify along with the intensity of the national audit investigation [24]. However, previous studies have only preliminarily tested the fiscal supervision efficiency of national auditing from the perspective of fiscal security and budget management; there is a lack of direct research centered on efficiency.

Scholars have further explored the relationship between national auditing and fiscal expenditure efficiency based on the functions of disclosing, punishment, and suggestion to find that synergy between these functions does improve local fiscal expenditure efficiency [18]. Scholars have also discovered that coordination across the disclosing, resisting, and preventing functions of national auditing does not have a linear relationship to local fiscal expenditure efficiency but rather exerts an inverted U-shaped impact that rises first and then declines [17].

### 2.2. Information Transmission Function of National Audit

A national audit provides accountability information as a safeguard for public power. Audit reports or other public information regarding fiscal expenditure can reduce information asymmetry, improve fiscal transparency [19], and improve fiscal supervision effectiveness. This efficiency is regarded in the present work as the information transmission function of the national audit.

The discussion on the information transmission function of a national audit mainly focuses on the two aspects of improving financial transparency and strengthening su-

pervision synergy and efficiency by providing high-quality audit information. National auditing can expose "tricks" or corruption in budget-preparation processes. The disclosure of audit results can reduce unreasonable spending by creating fiscal transparency in local governments [25], minimizing deviation between budgets and final accounts [26], and exerting other positive governance effects [27,28].

National auditing also enhances supervision force and effectiveness. With the professional support of national audit institutions, the NPC can use the high-quality information obtained to improve its own ability to supervise budgets and final accounts [29,30]. Public announcement systems for audit results may create workable foundations for new supervision models as well as citizen participation strategies [31]. Public involvement intensifies media reports on corruption, thus enhancing the effectiveness of the audit [32]. There have been previous studies on the information transmission function of national audits, but whether national auditing stimulates other supervisory mechanisms through information transmission thus producing governance effects remains unclear.

*2.3. Governance Effect of Media Attention*

As an information intermediary, the media can reduce the cost of information acquisition for information demanders by spreading information widely and rapidly. The media alleviates information asymmetry and allows public opinion to play a supervisory role. Media reports on the use of local fiscal funds not only place government fiscal revenue and expenditure information under public supervision but also may trigger internal intervention due to the "reputation mechanism" [33], where relevant departments seek to correct irregularities and regulate the allocation and use of fiscal funds [34]. Media attention thus exerts a governance effect on government fiscal revenue and expenditure behavior. For example, media reports have a significant inhibitory effect on the budgets of "three official vehicles" of government departments [20]. Media reports on audit findings can improve the fiscal transparency of local governments [35]; however, the mediating effect of media attention on improving fiscal expenditure efficiency in the context of national auditing is not yet known.

In summary, previous studies provide a research basis regarding the improvements of national audits on local fiscal expenditure efficiency. However, few scholars have empirically tested the governance effect of national audit information transmission function, especially whether the disclosure of audit results can further improve local fiscal expenditure efficiency by stimulating media attention.

## 3. Theoretical Analysis and Research Hypothesis

*3.1. Spatial Dependence of Local Fiscal Expenditure Efficiency*

According to the theory of spatial econometrics, a certain economic phenomenon or attribute value in a regional unit is always related to the corresponding economic phenomenon or attribute value in its neighboring regional unit; that is, there is spatial dependence.

Under the encouragement of the performance evaluation with GDP as the core, because fiscal expenditure is conducive to promoting local GDP growth, local government officials will not only formulate fiscal expenditure strategies according to the regional economic characteristics but also form spatial interaction of fiscal expenditure strategies by learning and imitating the fiscal expenditure strategies and behaviors of neighboring regions in order to get promoted and make the efficiency of fiscal expenditure space-dependent.

Some studies have shown that the spatial interaction of such expenditure strategies of local governments leads to the spatial disturbance correlation of fiscal expenditure efficiency [36]. The fiscal expenditure efficiency of each province shows obvious spatial spillover effect and club aggregation characteristics [17], and the spatial dependence of local fiscal expenditure efficiency is still gradually increasing [37].

Based on the above analysis, this paper puts forward hypothesis 1 (H1):

**Hypothesis 1 (H1):** *There is space dependence on the efficiency of local fiscal expenditure.*

### 3.2. Direct Effect of National Audit on Improving Local Fiscal Expenditure Efficiency

According to budget contract theory, there are multiple principal–agent relationships in the local government's fiscal budget, which may be contractual, publicity, and fiduciary in nature. As the trustee, the local government accepts the entrustment of the client, makes rational use of fiscal funds, provides public resources and services, and is responsible for fiscal resource management. However, information asymmetry and incomplete contracts make it difficult for the principal of local government fiscal fund allocation and usage to implement effective supervision. The agent may observe unreasonable fiscal resource allocation, illegal use of fiscal funds, or fiscal agency problems such as waste, which reduce local fiscal expenditure efficiency. In essence, these problems are caused by a lack of supervision. Supervisory and power-balance mechanisms must be strengthened to prevent inappropriate usage of public resources and to protect public property rights.

The national audit is an independent supervisory system of governance over checks and balances that works in accordance with public power [4] to protect public fiscal resources and property. By local governments in the field of finance, the national audit budget's allocation and execution shall be supervised along with funds use and management, and so on and so forth, and constraints placed on the fiscal funds use behavior of the local government to promote the rational use of fiscal funds, reduce the loss and waste of fiscal funds, improve the ability of fiscal capital operation and management, and improve the performance of use of public funds for local fiscal efficiency.

The national audit is an important part of the national political system and an institutional arrangement for the supervision and restriction of power by power according to law. The essence of the national audit is an endogenous "immune system", with the functions of disclosing, resisting, and preventing in the large system of national governance, in which disclosing is the foundation, resisting is the focus, and preventing is the purpose [4].

The national audit can reveal illegal behavior on the part of auditees, waste of public funds, unreasonable use of resources, defects in fiscal expenditure systems, obstacles or loopholes in management, and other issues; it promotes standardized usage of fiscal funds, ameliorates mismanagement of fiscal resources, and improves the efficiency of local government spending. The national audit plays the function of resisting, using legal power to identify and punish violations. National audits regulate behavior surrounding budgetary revenues and expenditures, ensure effective investment and usage of fiscal funds, increase outputs under given levels of fiscal capital investment, and ultimately improve the efficiency of fiscal expenditure.

The national audit also plays a preventive role, giving early warnings to the general public regarding emerging irregularities in audited units. Auditing may prevent corrupt intentions in audited units from turning into violations and further may prevent local problems from turning into global problems [38]. Targeted audit suggestions promote local governments to improve systems, standardize mechanisms, prevent risks and hidden dangers in local fiscal expenditure, and thus improve local fiscal expenditure efficiency.

Based on the above analysis, this paper puts forward hypothesis 2 (H2):

**Hypothesis 2a (H2a):** *Fiscal expenditure efficiency increases as the national audit disclosing function increases.*

**Hypothesis 2b (H2b):** *Fiscal expenditure efficiency increases as the national audit resisting function increases.*

**Hypothesis 2c (H2c):** *Fiscal expenditure efficiency increases as the national audit preventing function increases.*

### 3.3. Mediating Effect of Media Attention

Public opinion supervision can restrain local government behavior to a certain extent. Public involvement encourages agents to consider public interest in terms of fiscal expenditure, to optimize the allocation of public resources, and to improve the local public service level. By increasing the reporting intensity on local government fiscal expenditure, media

attention places public opinion pressure on fiscal fund users, promotes them to regulate the use of fiscal funds, and thus enhances the effectiveness of public supervision. However, as they are restricted by information release and access authority, most media cannot easily obtain information on the use of local government fiscal funds, which weakens said effectiveness.

National audit results may be released publicly by the media to report fiscal revenue and expenditure compliance and performance [32]. As the results are only source of information by which the local government announces to the public through verification in accordance with the information, the use of public funds for the media provides access to local government fiscal funds to use credible sources of information, to reduce the risk of the cost of information collection and that reported in the press, and enhance the possibility and willingness of media to pay attention to the use of local fiscal funds [18,32]. The media, to this effect, plays a role in the national audit. Audit proclamation of local government fiscal information quality is higher, the content is full, and the media information can be collected incrementally for its choice, confirming any widespread problem and repackaging audit results of public funding, involving the easier usage that has been the subject of media coverage, resulting in increased media attention to usage of fiscal capital.

Media attention can expose government fiscal spending violations and create enthusiasm amidst the public for "participatory governance", prompting local governments to respond to problems, thus promoting the standardized use of local fiscal funds and improving local fiscal expenditure efficiency.

Media attention functions via a constraint mechanism and a reputation mechanism. Media attention and other social accountability mechanisms may subject local governments to administrative punishment, which negatively affects the official careers and political futures of the involved personnel [39], thus forming a constraint mechanism. When media attention is given to fiscal irregularities, local governments may be motivated to avoid administrative punishment by regulating the use of fiscal funds, eliminating waste, and improving the efficiency of local fiscal expenditure.

Based on the reputation mechanism perspective, media attention, especially when there are consistent reports on fiscal irregularities, can degrade public trust in relevant units and departments, thus damaging their reputation. In efforts to rebuild (or protect) their image, these units will actively regulate the use of fiscal funds, minimize waste, and improve local fiscal expenditure efficiency.

In summary, the disclosure of national audit results has an information transmission function, causing media attention to the use of local government fiscal funds. Media attention will further promote the standardized use of local government fiscal funds, thus improving local fiscal expenditure efficiency. Media attention plays an intermediary role in the process of national auditing to improve local fiscal expenditure efficiency.

Based on the above analysis, this paper proposes hypothesis 3 (H3):

**Hypothesis 3 (H3):** *Media attention plays an intermediary role in the process of national auditing to improve local fiscal expenditure efficiency.*

## 4. Research Design

### 4.1. Sample Selection and Data Sources

China has 23 provinces, 5 autonomous regions, 4 municipalities directly under the Central Government, and 2 special administrative regions. Due to the different economic systems of Taiwan Province, Hong Kong, and Macao from the mainland and the lack of data from the Tibet Autonomous Region, we selected 30 provinces, autonomous regions, and municipalities directly under the central government in the Chinese mainland from 2007 to 2018 as samples. Data support includes the amount of illegal funds detected by national audits, the amount of problems dealt with by auditing organizations, and the audit suggestions put forward based on the China Audit Yearbook. The input data used to measure local fiscal expenditure efficiency come from the China Fiscal Yearbook, and the output data come from the official website of the National Bureau of Statistics. Media

attention is determined by the number of reports retrieved from *China's Major Newspapers Full-text Database* (CNKI) according to names of provinces, autonomous regions, and municipalities directly defined under the central government and "finance" as a keyword in the corresponding years. Fiscal transparency data come from the annual "China Fiscal Transparency Report" released by Shanghai University of Finance and Economics. The institutional environment data come from Wang Xiaolu et al.'s report on market index by provinces in China (2021). Other data come from the China Statistical Yearbook or the official website of the National Bureau of Statistics.

*4.2. Model Description*

In view of the possibility of spatial dependence on the efficiency of local fiscal expenditure of provinces and municipalities in China, we introduced spatial econometrics to analyze the mechanism of national audit to improve the efficiency of local fiscal expenditure. The explained variable is fiscal expenditure efficiency, the value of which falls into the interval of [0, 1]. To prevent any estimation parameter bias caused by OLS regression, we used a spatial Tobit model in this paper for estimation.

The general Tobit regression model of local fiscal expenditure efficiency without considering spatial factors is as follows:

$$y = X\beta + \varepsilon \tag{1}$$

where y is the dependent variable column vector of order $N \times 1$; X is the exogenous explanatory variable matrix of order $N \times K$, $\beta$ is the parameter vector to be estimated, and $\varepsilon$ is a random error term vector.

We next established a Tobit spatial lag (SAR) model to test the effect of fiscal expenditure efficiency in neighboring areas on fiscal expenditure efficiency in local areas. The external spatial lag variable reflects the spatial dependence of dependent variables. The Tobit spatial error (SEM) model measures the impact of error of the dependent variable in the neighboring region on the observed value of the local region. The spatial dependence of the model is reflected in the spatial disturbance term.

The SAR model of local fiscal expenditure efficiency is as follows:

$$y = \rho W y + X\beta + \varepsilon \tag{2}$$

where W is the spatial weight matrix of order $N \times N$, Wy is the spatial lag dependent variable, and $\rho$ is the spatial regression coefficient reflecting the spatial dependence relationship in the sample observation value (that is, the direction and degree of influence of the observation value Wy in the adjacent area on the observation value Y in the local area). We only examined whether there is spatial autocorrelation between the local fiscal expenditure efficiencies of geographically adjacent provinces, autonomous regions, and municipalities, so the spatial weight matrix W was set according to whether two provinces have a common boundary. If the i-th province and the j-th province have a common boundary, the $w_{ij}$ of the spatial weight coefficient of the two is 1; otherwise, it is 0.

The SEM model of local fiscal expenditure efficiency is as follows:

$$y = X\beta + \varepsilon \tag{3}$$

$$\varepsilon = \lambda W \varepsilon + \mu \tag{4}$$

where $\lambda$ is the spatial error coefficient, and $\mu$ is the random error vector of normal distribution. The parameter $\beta$ reflects the influence of independent variable X on dependent variable y, and the parameter $\lambda$ measures the influence of error impact on the observed value in the neighboring area.

### 4.3. Description of Variables

#### 4.3.1. Dependent Variable: Local Fiscal Expenditure Efficiency

We used the CCR model of data envelopment analysis to measure the fiscal expenditure efficiency of each province in each year. The input index is per capita fiscal expenditure value (CNY/person), which represents the fiscal resources consumed by each province to provide public services. Output indicators were selected with reference to the results of Zheng Shiqiao, Liang Siyuan [40], and Liu Jianghui and Wang Gongyu et al. [41]. Six types of first-level output indicators were selected, including infrastructure, medical and health care, public cultural services, education, ecological and environmental protection, and social security. Each first-level output indicator contains several second-level indicators (Table 1). The total output index value was calculated as the arithmetic average of the standard value of each secondary index as the corresponding first-level index value, then calculating the first index value.

**Table 1.** Output index table.

| First-Level Indicators | Second-Level Indicators |
|---|---|
| Infrastructure | Road area per capita (square meters) |
| | Post and telecommunications traffic per capita (CNY) |
| | Per capita electricity consumption (KWH) |
| | Buses per ten thousand people |
| Medical and health care | Number of medical beds per thousand people |
| | Number of health technicians per thousand people |
| Public cultural services | Number of library books per thousand people |
| | Number of theaters and cinemas per thousand people (number) |
| Education | Number of teachers in regular institutions of higher learning per thousand people |
| | Number of general secondary school teachers per thousand people |
| | Number of primary school teachers per thousand people |
| Ecological and environmental protection | Urban per capita green space (square meters) |
| | Comprehensive utilization rate of industrial solid waste |
| | Urban sewage treatment rate |
| | Harmless treatment rate of household garbage |
| Social security | Participation rate of basic endowment insurance |
| | Basic medical insurance participation rate |
| | Unemployment insurance participation rate |

#### 4.3.2. Main Explanatory Variable: National Audit

We refer to Wang Chunfei and Guo Yunnan [42] and Han Feng [43] in measuring the governance functions of the national audit according to disclosure, defense, and prevention. The natural logarithm of the ratio of the amount of money found by the audit institutions of each province to the fiscal expenditure of the local government was used as the proxy variable of the national audit disclosure function (NauditDis). The natural logarithm of cases transferred by audit institutions was used as the proxy variable of the national audit defense function (NauditRes). The natural logarithm of adopted audit recommendations was used as a proxy variable for the national audit prevention function (NauditPre).

#### 4.3.3. Mediating Variable: Media Attention

The disclosure of national audit results will trigger media attention on the use of fiscal funds. Public governance theory and citizen participation theory show that, with the

continuous development of society, the public and the media are increasingly demanding to know the use of government fiscal funds. Therefore, referring to the practice of Chi Guohua [44], this paper uses the names of provinces, autonomous regions, and municipalities directly under the central government and "finance" as keywords in the full-text database of important newspapers in China (CNKI) to conduct title retrieval according to the corresponding years so as to obtain the number of media reports and measure the degree of media attention. The reason for title retrieval in this paper is that, compared with the word "finance" only appearing in the full text, media reports containing the word "finance" in the title are more eye-catching, resulting in greater social influence than the word "finance" only appearing in the full text.

### 4.3.4. Control Variables

With reference to the literature, we selected the following variables that may affect local fiscal expenditure efficiency as control variables:

(1)　Fiscal transparency (FisTrans)

Fiscal transparency is measured by the "Fiscal Transparency Index" in the annual "Chinese Fiscal Transparency Report" published by Shanghai University of Finance and Economics. Generally speaking, higher fiscal transparency indicates more standardized the use of local government funds and more efficient fiscal expenditure. The expected sign is positive.

(2)　Institutional environment (Market)

We adopt the total marketization index in Marketization Index by Provinces in China (Wang Xiaolu et al., 2021) to measure the institutional environment of each province. A stronger institutional environment in a given a region generally indicates better governance, including more efficient local fiscal expenditure. The expected sign is positive.

(3)　Population density (PopDens)

We measured population density by population per square meter. Afonso believed that in areas with high population density, the government shows a scale effect in providing public services, which can improve the local fiscal expenditure efficiency [45]. The expected coefficient sign is positive.

(4)　Education level of residents (HR)

Referring to Peng Guohua [46], we used the rate of return on education investment to convert the education years of residents into human capital so as to measure education levels in our sample. More-educated citizens are expected to be more aware of supervisory mechanisms, which would create pressure that reduces the illegal use of funds by local governments and improves local fiscal expenditure efficiency. The expected sign is positive.

(5)　Fiscal independence (FD)

We used the proportion of fiscal revenue to fiscal expenditure to measure fiscal independence. When the degree of fiscal autonomy of a region is improved, the local government can independently control more funds, thus increasing the possibility of illegal fiscal behavior and reducing the local fiscal expenditure efficiency. The expected sign is negative.

(6)　Population structure (PopStr)

We measured the population structure as the birth rate minus the death rate in our sample. A younger-skewing population structure indicates a strong labor force, thus improving local fiscal expenditure efficiency. The expected sign is positive.

(7)　Industrial structure (IndStr)

We measured the industrial structure as the value added of the tertiary industry/regional GDP. A strong industrial structure is conducive to local economic development, thus improving local fiscal expenditure efficiency. The expected sign is positive.

Each variable and its definition are shown in Table 2:

**Table 2.** Variable definitions.

| Variable Types | Name | Variable Description | Variable Definitions |
|---|---|---|---|
| Dependent variable | UseEff | Local fiscal expenditure efficiency | Data envelopment analysis CCR model used for measurement. |
| Main explanatory variable | NauditDis | National audit disclosing function | Find natural logarithm of amount in question relative to total government public expenditure |
| | NauditRes | National audit resisting function | Natural number of cases referred by auditors. |
| | NauditPre | National audit preventing function | The natural number of audit recommendations adopted. |
| Mediating variable | Media | Media attention | The number of media reports obtained from China National Knowledge Network (CNKI) *Full-text Database of Major Newspapers in China* by corresponding year, using the names of provinces, autonomous regions, and municipalities directly under the central government and "finance" as keywords. |
| Control variables | FisTrans | Fiscal transparency | It is measured by the "Fiscal Transparency Index" in the annual China Fiscal Transparency Report published by Shanghai University of Finance and Economics. |
| | Market | Institutional environment | Measure with market aggregate index. |
| | PopDens | Population density | Population per square meter. |
| | HR | Education level of residents | The return on investment in education is used to convert the number of years residents have been educated. |
| | FD | Fiscal independence | The ratio of fiscal revenue to fiscal expenditure. |
| | PopStr | Population structure | The difference between birth and death rates. |
| | IndStr | Industrial structure | Tertiary industry added value divided by regional GDP. |

## 5. Empirical Analysis

### 5.1. Descriptive Statistical Analysis

Table 3 shows our descriptive statistical analysis of the main variables described above. The average value of local fiscal expenditure efficiency (UseEff) is 0.674, the minimum value is 0.298, and the maximum value is 1, indicating that local fiscal expenditure efficiency is generally low and that there are significant differences among different provinces. The disclosing function (NauditDis), resisting function (NauditRes), and preventing function (NauditPre) of the national audits also differ considerably among provinces. The mean value of media attention (Media) is 16.561, indicating that the media of each province gives a reasonable amount of attention to the use of fiscal government funds. However, the minimum value is 1, and the maximum value is 58, indicating substantial differences in the media reports of each province.

To test the collinearity among variables, we calculated the VIF values of each variable to find that all are less than 10, indicating no high collinearity among variables.

**Table 3.** Descriptive statistical analysis.

| Variable | Obs | Mean | Std. Dev. | Min | Max |
|---|---|---|---|---|---|
| UseEff | 360 | 0.674 | 0.161 | 0.298 | 1.000 |
| NauditDis | 360 | −1.014 | 0.643 | −2.948 | 1.140 |
| NauditRes | 360 | 4.664 | 1.339 | 0.000 | 7.596 |
| NauditPre | 360 | 8.366 | 0.992 | 5.509 | 10.234 |
| Media | 360 | 16.561 | 10.295 | 1.000 | 58.000 |
| FisTrans | 360 | 36.795 | 15.571 | 14.000 | 77.700 |
| Market | 360 | 6.280 | 1.817 | 2.330 | 10.388 |
| PopDens | 360 | 0.046 | 0.068 | 0.0007 | 0.393 |
| HR | 360 | 1.467 | 0.129 | 1.182 | 1.975 |
| FD | 360 | 0.511 | 0.195 | 0.148 | 0.951 |
| PopStr | 360 | 5.275 | 2.660 | −1.000 | 11.780 |
| IndStr | 360 | 0.462 | 0.091 | 0.298 | 0.831 |

*5.2. Basic Regression Analysis*

There is a certain degree of inter-governmental competition in China. Local governments tend to imitate their "neighbors" in terms of fiscal expenditure, which leads to spatial dependence in local fiscal expenditure efficiency across provinces. We tested whether local fiscal expenditure efficiency is spatially dependent by the Moran's I index and Geary's C index, then decided whether to adopt the spatial econometric model according to the results.

As shown in Table 4, both the Moran's I index and Geary's C index of local fiscal expenditure efficiency are significantly positive. This shows that there is a spatial dependence on the efficiency of local fiscal expenditure, and hypothesis 1 is confirmed. Therefore, we need to incorporate spatial geographic factors into the regression analysis model and use spatial econometrics methods to investigate. We used a Tobit spatial lag model and Tobit spatial error model to estimate and test our parameters, then investigated the results of LM Lag and LM Err tests. We found that the LM Err statistic of local fiscal expenditure efficiency (UseEff) is significant at the 1% level, while the LM Lag statistic was not. Thus, the regression results should be analyzed by a spatial error model.

**Table 4.** Moran's I index and Geary's C index test of local fiscal expenditure efficiency.

| Year | Moran's I | | | | | Geary's C | | | | |
|---|---|---|---|---|---|---|---|---|---|---|
| | I | E(I) | Sd(I) | Z | *p*-Value | C | E(C) | Sd(C) | Z | *p*-Value |
| 2007 | 0.214 | −0.034 | 0.120 | 2.069 | 0.019 | 0.755 | 1.000 | 0.130 | −1.889 | 0.029 |
| 2008 | 0.350 | −0.034 | 0.121 | 3.171 | 0.001 | 0.652 | 1.000 | 0.127 | −2.741 | 0.003 |
| 2009 | 0.273 | −0.034 | 0.121 | 2.538 | 0.006 | 0.699 | 1.000 | 0.128 | −2.352 | 0.009 |
| 2010 | 0.368 | −0.034 | 0.121 | 3.323 | 0.000 | 0.662 | 1.000 | 0.128 | −2.652 | 0.004 |
| 2011 | 0.312 | −0.034 | 0.121 | 2.869 | 0.002 | 0.714 | 1.000 | 0.129 | −2.221 | 0.013 |
| 2012 | 0.296 | −0.034 | 0.120 | 2.744 | 0.003 | 0.735 | 1.000 | 0.129 | −2.049 | 0.020 |
| 2013 | 0.284 | −0.034 | 0.121 | 2.627 | 0.004 | 0.721 | 1.000 | 0.127 | −2.198 | 0.014 |
| 2014 | 0.239 | −0.034 | 0.121 | 2.256 | 0.004 | 0.754 | 1.000 | 0.127 | −1.943 | 0.026 |
| 2015 | 0.215 | −0.034 | 0.121 | 2.069 | 0.019 | 0.748 | 1.000 | 0.129 | −1.960 | 0.025 |
| 2016 | 0.176 | −0.034 | 0.121 | 1.733 | 0.042 | 0.795 | 1.000 | 0.127 | −1.612 | 0.053 |
| 2017 | 0.124 | −0.034 | 0.122 | 1.308 | 0.095 | 0.790 | 1.000 | 0.126 | −1.661 | 0.048 |
| 2018 | 0.166 | −0.034 | 0.120 | 1.665 | 0.048 | 0.794 | 1.000 | 0.129 | −1.590 | 0.056 |
| 2019 | 0.155 | −0.034 | 0.121 | 1.573 | 0.058 | 0.796 | 1.000 | 0.128 | −1.584 | 0.057 |

To verify the correlation between national auditing and local fiscal expenditure efficiency, we tested three functions of the national audit by spatial metrology. We delayed the national audit variable by one period to prevent any reverse causality problems between the national audit and local fiscal expenditure efficiency. The results are shown in columns (4) to (6) of Table 5. The regression coefficients of the three main explanatory variables (L.NauditDis, L.NauditRes, and L.NauditPre) are significantly positive at the 1% level, indicating that all three functions of the national audit are significantly positively correlated with local fiscal expenditure efficiency. Per the regression coefficient, local fiscal expenditure efficiency will increase by 0.051% on average if the disclosing function is increased by 1%. The resisting function increases by 1%, and the efficiency of local government expenditure increases by 0.019% on average. Prevention functions were increased by 1%, and the efficiency of local government expenditure increased by 0.037% on average. Thus, hypothesis 2 is verified.

**Table 5.** Basic regression analysis results.

| Variables | (1) UseEff | (2) UseEff | (3) UseEff | (4) UseEff | (5) UseEff | (6) UseEff |
|---|---|---|---|---|---|---|
| L.NauditDis (z) | 0.086 *** (7.54) | | | 0.051 *** (5.69) | | |
| L.NauditRes (z) | | 0.061 *** (8.81) | | z | 0.019 *** (3.52) | |
| L.NauditPre (z) | | | 0.077 *** (17.23) | | | 0.037 *** (4.93) |
| FisTrans (z) | | | | 0.001 *** (2.60) | 0.001 ** (2.54) | 0.001 *** (2.76) |
| Market (z) | | | | 0.040 *** (6.85) | 0.035 *** (5.46) | 0.036 *** (5.91) |
| PopDens (z) | | | | −0.389 *** (−3.46) | −0.368 *** (−3.12) | −0.413 *** (−3.63) |
| HR (z) | | | | −0.191 *** (−2.64) | −0.154 ** (−2.08) | −0.081 (−1.02) |
| FD (z) | | | | 0.259 *** (4.75) | 0.307 *** (5.43) | 0.271 *** (4.83) |
| PopStr (z) | | | | 0.001 (0.62) | 0.002 (0.87) | 0.003 (1.25) |
| IndStr (z) | | | | −0.900 *** (−10.51) | −0.871 *** (−9.70) | −0.820 *** (−9.07) |
| Constant (z) | 0.635 *** (32.56) | 0.292 *** (8.60) | 0.015 (0.64) | 0.961 *** (10.08) | 0.760 *** (7.77) | 0.429 *** (3.13) |
| Lambda (z) | 0.045 *** (9.38) | 0.068 *** (7.74) | 0.106 *** (9.13) | 0.014 *** (3.76) | 0.014 *** (3.07) | 0.018 ** (2.14) |
| Sigma (z) | 0.135 *** (26.82) | 0.128 *** (26.74) | 0.119 *** (26.26) | 0.098 *** (26.83) | 0.101 *** (26.83) | 0.099 *** (26.83) |
| LM-Lag (*p*-Value) | 1.553 (0.216) | 1.474 (0.225) | 1.314 (0.252) | 0.948 (0.330) | 0.888 (0.346) | 0.919 (0.337) |
| LM-Err (*p*-Value) | 163.774 *** (0.000) | 214.322 *** (0.000) | 93.381 *** (0.000) | 60.571 *** (0.000) | 49.737 *** (0.000) | 50.175 *** (0.000) |
| LR Test (*p*-Value) | 87.978 *** (0.000) | 59.858 *** (0.000) | 83.372 *** (0.000) | 14.161 *** (0.000) | 9.444 *** (0.002) | 4.598 ** (0.032) |
| Adjust-$R^2$ | 0.036 | 0.184 | 0.153 | 0.488 | 0.476 | 0.508 |
| Observations | 360 | 360 | 360 | 360 | 360 | 360 |

Note: ** significant at 5% level, and *** significant at 1% level.

The national audit, as a supervisory governance system, appears to have reduced waste in the use of government funding and improved local fiscal expenditure efficiency

throughout our sample. Auditees are punished when they misuse funds, defects (e.g., waste) in capital expenditures are reduced by auditing, and obstacles and loopholes in management rules are minimized after an audit is conducted.

The regression coefficient λ of the spatial error in local fiscal expenditure efficiency (UseEff) is significantly positive, indicating that local fiscal expenditure efficiency has spatial autocorrelation across our sample. This may be because GDP-oriented performance assessment during the observation period produced imitation effects in the formulation of local fiscal expenditure policies due to competition among local governments. Our LR test results show that this model cannot degenerate from the spatial error model to the OLS model, which rationalizes our spatial econometrics analysis.

*5.3. Mediating Effect Test of Media Attention*

We refer to the intermediary effect test method of Baron and Kenny (1986) and Wen and Ye (2014) [47,48] to test the mediating role of media attention in national auditing to improve local fiscal expenditure efficiency. We first conducted a regression between explained variables and explanatory variables as shown in columns (4)–(6) of Table 5. The regression coefficients of explanatory variables are significantly positive at the 1% level. Next, we conducted a regression with mediating variable media attention and explanatory variables as shown in columns (1), (3), and (5) of Table 6. The regression coefficient of the disclosing function (NauditDis) is significantly positive at the 1% level, that of the resisting function (NauditRes) is significantly positive at the 5% level, and that of the preventing function (NauditRes) is significantly positive at the 1% level. We next regressed the explanatory variables and intermediary variable as shown in columns (2), (4), and (6) of Table 6. The interpretation and mediation variables in the regression coefficients were significantly positive, indicating a partial mediation effect of media attention on the improvement local fiscal expenditure efficiency after national auditing.

**Table 6.** Mediating effect of media attention.

| Variables | (1) Media | (2) UseEff | (3) Media | (4) UseEff | (5) Media | (6) UseEff |
|---|---|---|---|---|---|---|
| L.NauditDis | 3.372 *** | 0.049 *** | | | | |
| (z) | (4.20) | (5.42) | | | | |
| L.NauditRes | | | 1.282 ** | 0.020 *** | | |
| (z) | | | (2.43) | (3.75) | | |
| L.NauditPre | | | | | 1.882 *** | 0.036 *** |
| (z) | | | | | (3.13) | (4.91) |
| Media | | 0.001 ** | | 0.002 *** | | 0.001 ** |
| (z) | | (1.99) | | (2.89) | | (2.51) |
| FisTrans | 0.041 | 0.001 *** | 0.058 | 0.001 *** | 0.059 | 0.001 *** |
| (z) | (1.00) | (2.78) | (1.39) | (2.65) | (1.42) | (2.94) |
| Market | 1.571 *** | 0.041 *** | 1.391 ** | 0.036 *** | 1.253 ** | 0.037 *** |
| (z) | (2.63) | (7.02) | (2.28) | (5.63) | (2.05) | (6.12) |
| PopDens | −3.420 | −0.401 *** | −2.312 | −0.376 *** | −3.974 | −0.423 *** |
| (z) | (−0.35) | (−3.58) | (−0.23) | (−3.23) | (−0.40) | (−3.76) |
| HR | 10.286 | −0.192 *** | 13.558 ** | −0.157 ** | 17.670 ** | −0.087 |
| (z) | (1.58) | (−2.68) | (2.03) | (−2.16) | (2.57) | (−1.13) |
| FD | 2.432 | 0.231 *** | 4.091 | 0.267 *** | 4.284 | 0.236 *** |
| (z) | (0.42) | (4.12) | (0.69) | (4.63) | (0.73) | (4.15) |
| PopStr | 1.001 *** | 0.001 | 1.064 *** | 0.001 | 1.089 *** | 0.002 |
| (z) | (4.96) | (0.24) | (5.15) | (0.32) | (5.29) | (0.76) |
| IndStr | −13.058 * | −0.873 *** | −8.738 | −0.826 *** | −9.466 | −0.787 *** |
| (z) | (−1.67) | (−10.13) | (−1.00) | (−9.16) | (−1.14) | (−8.70) |

**Table 6.** *Cont.*

| Variables | (1) | (2) | (3) | (4) | (5) | (6) |
|---|---|---|---|---|---|---|
| | **Media** | **UseEff** | **Media** | **UseEff** | **Media** | **UseEff** |
| Constant | −2.757 | 0.943 *** | −18.782 ** | 0.738 *** | −33.388 *** | 0.424 *** |
| (z) | (−0.33) | (9.98) | (−2.01) | (7.73) | (−2.90) | (3.22) |
| Lambda | | 0.013 *** | | 0.013 *** | | 0.017 ** |
| (z) | | (3.62) | | (2.83) | | (2.08) |
| Sigma | | 0.097 *** | | 0.099 *** | | 0.098 *** |
| (z) | | (26.83) | | (26.83) | | (26.83) |
| LM-Lag | | 1.038 | | 1.026 | | 1.033 |
| (*p*-Value) | | (0.308) | | (0.311) | | (0.309) |
| LM-Err | | 62.702 *** | | 55.916 *** | | 56.506 *** |
| (*p*-Value) | | (0.000) | | (0.000) | | (0.000) |
| LR Test | | 13.139 *** | | 8.002 *** | | 4.307 ** |
| (*p*-Value) | | (0.000) | | (0.005) | | (0.038) |
| Adjust-$R^2$ | 0.332 | 0.495 | 0.309 | 0.492 | 0.316 | 0.586 |
| Observations | 360 | 360 | 360 | 360 | 360 | 360 |

Note: * significant at 10% level, ** significant at 5% level, and *** significant at 1% level.

The mediating effect of media attention indicates that a national audit provides high-quality information for numerous supervisory subjects through the disclosure of audit results on the basis of the disclosing, resisting, and preventing functions. More effective national audit functions result in disclosure of higher-quality local government fiscal information. Information of higher quality has more issues available for the media to select, confirm, and repackage, thus gaining more public attention. Media attention thus enhances the supervisory effect of public opinion, further promotes the standardized use of funds, and ultimately promotes local fiscal expenditure efficiency.

*5.4. Robustness Test*

In order to enhance the reliability of our findings, we conducted a robustness test by replacing the core explanatory variables. The natural logarithm of the amount of detected problems per unit was used as a proxy variable of the disclosing function (NauditDis), and the natural logarithm of the amount of transferred problems per unit was used as a proxy variable of the resisting function (NauditRes). The natural logarithm of the number of audit recommendations was used as a proxy variable of the preventing function (NauditPre). The regression results are shown in Table 7. The regression coefficient between the national audit disclosure function and prevention function and local fiscal expenditure efficiency is significantly positive, indicating that our conclusion is robust.

**Table 7.** Robustness test (I).

| Variables | (1) | (2) | (3) |
|---|---|---|---|
| | **UseEff** | **UseEff** | **UseEff** |
| L.NauditDis | 0.019 ** | | |
| (z) | (2.27) | | |
| L.NauditRes | | 0.009 | |
| (z) | | (1.43) | |
| L.NauditPre | | | 0.059 *** |
| (z) | | | (5.41) |
| FisTrans | 0.001 *** | 0.002 *** | 0.001 ** |
| (z) | (3.27) | (3.56) | (2.36) |
| Market | 0.039 *** | 0.040 *** | 0.039 *** |
| (z) | (6.20) | (6.26) | (5.59) |

**Table 7.** *Cont.*

| Variables | (1) | (2) | (3) |
|---|---|---|---|
| | UseEff | UseEff | UseEff |
| PopDens | −0.465 *** | −0.441 *** | −0.440 *** |
| (z) | (−4.00) | (−3.77) | (−3.82) |
| HR | −0.212 *** | −0.174 ** | −0.007 |
| (z) | (−2.72) | (−2.32) | (−0.13) |
| FD | 0.279 *** | 0.285 *** | 0.194 ** |
| (z) | (4.93) | (5.01) | (2.49) |
| PopStr | 0.001 | 0.002 | 0.004 |
| (z) | (0.51) | (0.75) | (1.59) |
| IndStr | −0.973 *** | −0.963 *** | −0.719 *** |
| (z) | (−10.78) | (−10.58) | (−6.73) |
| Constant | 0.807 *** | 0.836 *** | 0.095 |
| (z) | (8.27) | (8.70) | (0.78) |
| Lambda | 0.017 *** | 0.015 *** | 0.051 * |
| (z) | (3.76) | (3.52) | (1.69) |
| Sigma | 0.102 *** | 0.102 *** | 0.097 *** |
| (z) | (26.83) | (26.83) | (26.30) |
| LM-Lag | 0.793 | 0.825 | 0.987 |
| (*p*-Value) | (0.373) | (0.364) | (0.344) |
| LM-Err | 49.882 *** | 46.390 *** | 49.159 *** |
| (*p*-Value) | (0.000) | (0.000) | (0.000) |
| LR Test | 14.172 *** | 12.381 *** | 2.863 * |
| (*p*-Value) | (0.000) | (0.000) | (0.091) |
| Adjusted-$R^2$ | 0.449 | 0.454 | 0.529 |
| Observations | 360 | 360 | 360 |

Note: * significant at 10% level, ** significant at 5% level, and *** significant at 1% level.

We also used a traditional year fixed-effect test and individual-year bidirectional fixed effect test to examine the reliability of our findings. The results are shown in Table 8. The disclosing, resisting, and preventing functions are still significant after testing fixed effects. The bidirectional fixed effect test shows that the disclosing function is still significant, while disclosing, resisting, and preventing are not significant though still positively correlated. The slightly less-than-ideal bidirectional fixed effect test results may be attributable to our relatively large quantity of control variables and relatively brief sample period.

**Table 8.** Robustness test (II).

| Variables | (1) | (2) | (3) | (4) | (5) | (6) |
|---|---|---|---|---|---|---|
| | UseEff | UseEff | UseEff | UseEff | UseEff | UseEff |
| L.NauditDis | 0.063 *** | | | 0.027 *** | | |
| (t) | (6.68) | | | (4.27) | | |
| L.NauditRes | | 0.039 *** | | | 0.00009 | |
| (t) | | (6.34) | | | (0.02) | |
| L.NauditPre | | | 0.050 *** | | | 0.002 |
| (t) | | | (7.26) | | | (0.18) |
| FisTrans | 0.002 *** | 0.002 *** | 0.002 *** | 0.001 *** | 0.001 *** | 0.001 *** |
| (t) | (3.48) | (3.92) | (4.11) | (2.80) | (3.18) | (3.19) |

**Table 8.** *Cont.*

| Variables | (1) | (2) | (3) | (4) | (5) | (6) |
|---|---|---|---|---|---|---|
| | UseEff | UseEff | UseEff | UseEff | UseEff | UseEff |
| Market | 0.056 *** | 0.051 *** | 0.047 *** | −0.010 | −0.011 | −0.011 |
| (t) | (7.92) | (7.15) | (6.77) | (−1.34) | (−1.54) | (−1.53) |
| PopDens | −0.493 *** | −0.429 *** | −0.487 *** | 0.062 | −0.317 | −0.316 |
| (t) | (−4.29) | (−3.68) | (−4.28) | (0.10) | (−0.50) | (−0.50) |
| HR | −0.171 ** | −0.082 | 0.019 | −0.111 | −0.043 | −0.045 |
| (t) | (−2.24) | (−1.05) | (0.24) | (−0.86) | (−0.32) | (−0.34) |
| FD | 0.128 * | 0.132 * | 0.146 ** | −0.474 *** | −0.503 *** | −0.503 *** |
| (t) | (1.87) | (1.93) | (2.17) | (−5.11) | (−5.28) | (−5.29) |
| PopStr | 0.002 | 0.004 | 0.004 * | −0.006 | −0.005 | −0.005 |
| (t) | (0.82) | (1.57) | (1.79) | (−1.52) | (−1.17) | (−1.17) |
| IndStr | −0.911 *** | −0.722 *** | −0.775 *** | 0.186 | 0.201 | 0.205 |
| (t) | (−9.97) | (−7.13) | (−8.17) | (1.33) | (1.40) | (1.41) |
| Constant | 0.984 *** | 0.562 *** | 0.212 | 1.106 *** | 1.015 *** | 1.004 *** |
| (t) | (10.16) | (5.18) | (1.60) | (3.74) | (3.33) | (3.25) |
| Year FE | Yes | Yes | Yes | Yes | Yes | Yes |
| Individual FE | | | | Yes | Yes | Yes |
| Adjusted-$R^2$ | 0.647 | 0.643 | 0.655 | 0.911 | 0.906 | 0.906 |
| Observations | 360 | 360 | 360 | 360 | 360 | 360 |

Note: * significant at 10% level, ** significant at 5% level, and *** significant at 1% level.

To control for any endogenous problems caused by national auditing and local fiscal expenditure efficiency, we used the spatial panel GMM estimation method to re-estimate the original equation based on the original spatial Tobit model. The results are shown in Table 9. The regression coefficient of the national audit disclosing function (NauditDis) and local fiscal output efficiency (UseEff) is significantly positive at the 1% level, and that of the preventing function (NauditPre) and local fiscal output efficiency (UseEff) is significantly positive at the 10% level, indicating that after controlling for endogeneity, our conclusions remain robust.

**Table 9.** Robustness test (III).

| Variables | (1) | (2) | (3) |
|---|---|---|---|
| | UseEff | UseEff | UseEff |
| L.NauditDis | 0.028 *** | | |
| (t) | (3.93) | | |
| L.NauditRes | | 0.003 | |
| (t) | | (0.51) | |
| L.NauditPre | | | 0.018 * |
| (t) | | | (1.93) |
| FisTrans | 0.001 ** | 0.001 ** | 0.001 ** |
| (t) | (2.10) | (2.58) | (2.48) |
| Market | 0.012 * | 0.013 * | 0.012 * |
| (t) | (1.74) | (1.71) | (1.70) |
| PopDens | 0.108 | 0.008 | 0.0003 |
| (t) | (0.36) | (0.03) | (0.00) |
| HR | −0.351 *** | −0.290 *** | −0.332 *** |
| (t) | (−4.38) | (−3.53) | (−3.96) |

**Table 9.** *Cont.*

| Variables | (1) | (2) | (3) |
|---|---|---|---|
| | UseEff | UseEff | UseEff |
| FD | 0.022 | 0.036 | 0.055 |
| (t) | (0.28) | (0.44) | (0.67) |
| PopStr | −0.014 *** | −0.013 *** | −0.013 *** |
| (t) | (−3.65) | (−3.55) | (−3.52) |
| IndStr | −0.426 *** | −0.448 *** | −0.388 *** |
| (t) | (−3.28) | (−3.31) | (−2.91) |
| Constant | 1.370 *** | 1.232 *** | 1.124 *** |
| (t) | (12.46) | (11.44) | (9.42) |
| Adjusted-$R^2$ | 0.297 | 0.272 | 0.323 |
| Observations | 360 | 360 | 360 |

Note: * significant at 10% level, ** significant at 5% level, and *** significant at 1% level.

Central inspection work is an important inner-party supervision system responsible for decision implementation as well as arrangements of the central committee of the Communist Party of China (CPC), standardizing local government behavior and eliminating corruption. Since the 18th CPC National Congress, the central government has conducted 19 rounds of inspections of provincial (autonomous region and municipality), local, central, and state organizations alongside key enterprises. To control for any endogenous problems caused by the omission of central inspection, we added the control variable of central inspection work (CIK). CIK is assigned to 1 if there is a central patrol in a province and 0 otherwise. The results are shown in Table 10. The regression coefficient between the three functions of the national audit and local fiscal expenditure efficiency (UseEff) is still significantly positive after adding central inspection control, indicating that our conclusions are still robust.

**Table 10.** Robustness test (IV).

| Variables | (1) | (2) | (3) |
|---|---|---|---|
| | UseEff | UseEff | UseEff |
| L.NauditDis | 0.051 *** | | |
| (z) | (5.70) | | |
| L.NauditRes | | 0.020 *** | |
| (z) | | (3.58) | |
| L.NauditPre | | | 0.037 *** |
| (z) | | | (4.94) |
| CIK | 0.010 | 0.013 | 0.009 |
| (z) | (0.66) | (0.86) | (0.60) |
| FisTrans | 0.001 ** | 0.001 ** | 0.001 *** |
| (z) | (2.57) | (2.49) | (2.74) |
| Market | 0.040 *** | 0.035 *** | 0.036 *** |
| (z) | (6.87) | (5.46) | (5.93) |
| PopDens | −0.389 *** | −0.366 *** | −0.413 *** |
| (z) | (−3.46) | (−3.12) | (−3.63) |
| HR | −0.193 *** | −0.156 ** | −0.082 |
| (z) | (−2.66) | (−2.11) | (−1.04) |
| FD | 0.258 *** | 0.307 *** | 0.270 *** |
| (z) | (4.74) | (5.43) | (4.84) |

**Table 10.** *Cont.*

| Variables | (1) | (2) | (3) |
|---|---|---|---|
| | UseEff | UseEff | UseEff |
| PopStr | 0.001 | 0.002 | 0.003 |
| (z) | (0.58) | (0.82) | (1.21) |
| IndStr | −0.898 *** | −0.867 *** | −0.818 *** |
| (z) | (−10.48) | (−9.64) | (−9.05) |
| Constant | 0.962 *** | 0.759 *** | 0.430 *** |
| (z) | (10.10) | (7.79) | (3.16) |
| Lambda | 0.013 *** | 0.014 *** | 0.018 ** |
| (z) | (3.73) | (3.02) | (2.13) |
| Sigma | 0.098 *** | 0.100 *** | 0.099 *** |
| (z) | (26.83) | (26.83) | (26.83) |
| LM-Lag | 0.918 | 0.850 | 0.889 |
| (*p*-Value) | (0.338) | (0.357) | (0.346) |
| LM-Err | 58.7848 *** | 47.946 *** | 49.358 *** |
| (*p*-Value) | (0.000) | (0.000) | (0.000) |
| LR Test | 13.921 *** | 9.125 *** | 4.541 ** |
| (*p*-Value) | (0.000) | (0.002) | (0.033) |
| Adjusted-$R^2$ | 0.488 | 0.477 | 0.508 |
| Observations | 360 | 360 | 360 |

Note: ** significant at 5% level, and *** significant at 1% level.

We also narrowed the sample size to further investigate the validity of our results. After the global fiscal crisis caused by the U.S. subprime mortgage crisis in 2008, the Chinese government launched a CNY 4 trillion fiscal stimulus plan to promote economic growth. The investment of provinces and cities then increased substantially, and the fiscal expenditure behavior and amount of local government funding showed some unusual changes. To eliminate the impact of this plan on the fiscal expenditure efficiency of local governments, we excluded the data from 2007 to 2009 and performed regression on the reduced sample according to the original method. The results are shown in Table 11. The regression coefficients of the three functions of the national audit and local fiscal expenditure efficiency (UseEff) are significantly positive, indicating that our main conclusions are still valid.

**Table 11.** Robustness test (V).

| Variables | (1) | (2) | (3) |
|---|---|---|---|
| | UseEff | UseEff | UseEff |
| L.NauditDis | 0.060 *** | | |
| (z) | (6.15) | | |
| L.NauditRes | | 0.027 *** | |
| (z) | | (4.37) | |
| L.NauditPre | | | 0.045 *** |
| (z) | | | (5.37) |
| FisTrans | 0.002 *** | 0.002 *** | 0.002 *** |
| (z) | (3.70) | (3.47) | (4.46) |
| Market | 0.035 *** | 0.026 *** | 0.031 *** |
| (z) | (5.35) | (3.64) | (4.58) |
| PopDens | −0.384 *** | −0.375 *** | −0.446 *** |
| (z) | (−3.03) | (−2.83) | (−3.50) |

**Table 11.** *Cont.*

| Variables | (1) | (2) | (3) |
|---|---|---|---|
| | UseEff | UseEff | UseEff |
| HR | −0.241 *** | −0.200 ** | −0.117 |
| (z) | (−2.77) | (−2.23) | (−1.30) |
| FD | 0.317 *** | 0.403 *** | 0.334 *** |
| (z) | (4.62) | (5.65) | (4.73) |
| PopStr | −0.002 | −0.001 | −0.001 |
| (z) | (−0.70) | (−0.53) | (−0.40) |
| IndStr | −0.883 *** | −0.827 *** | −0.763 *** |
| (z) | (−8.52) | (−7.66) | (−7.00) |
| Constant | 1.030 *** | 0.775 *** | 0.372 ** |
| (z) | (9.23) | (6.76) | (2.53) |
| Lambda | 0.011 *** | 0.010 ** | 0.015 |
| (z) | (2.95) | (1.98) | (1.55) |
| Sigma | 0.096 *** | 0.099 *** | 0.098 *** |
| (z) | (23.24) | (23.24) | (23.24) |
| LM-Lag | 0.478 | 0.483 | 0.431 |
| (*p*-Value) | (0.489) | (0.487) | (0.512) |
| LM-Err | 25.001 *** | 27.296 *** | 19.458 *** |
| (*p*-Value) | (0.000) | (0.000) | (0.000) |
| LR Test | 8.729 *** | 3.293 ** | 2.387 |
| (*p*-Value) | (0.003) | (0.048) | (0.122) |
| Adjusted-$R^2$ | 0.514 | 0.509 | 0.536 |
| Observations | 270 | 270 | 270 |

Note: ** significant at 5% level, and *** significant at 1% level.

*5.5. Coupling Effect Test*

The national audit is a subsystem of national governance and is consistent with other state supervisory systems. The national audit influences and adapts to the political, economic, cultural, social, and institutional environments surrounding it while promoting organic coordination among various types of supervisory systems. Therefore, there is a coupling effect between national auditing and media attention.

With reference to Peng Chong [17], we calculated the decoupling and coordination degree (Co-Nauditdis, Co-Nauditres, and Co-Nauditpre) between national audit disclosing, resisting, and preventing functions and media attention; we tested the correlation between these variables and local fiscal expenditure efficiency as shown in Table 12. The regression coefficients of the coupling degree (Co-Nauditdis and Co-Nauditpre) between the national audit disclosure function, prevention function, and media attention and local fiscal expenditure efficiency (UseEff) are significantly positive, indicating that national audit and media attention coupling improves local fiscal expenditure efficiency.

**Table 12.** Coupling effect test.

| Variables | (1) | (2) | (3) |
|---|---|---|---|
| | UseEff | UseEff | UseEff |
| L.Co-NauditDis | 0.078 ** | | |
| (z) | (2.04) | | |
| L.Co-NauditRes | | 0.063 | |
| (z) | | (1.44) | |

**Table 12.** *Cont.*

| Variables | (1) | (2) | (3) |
|---|---|---|---|
| | **UseEff** | **UseEff** | **UseEff** |
| L.Co-NauditPre | | | 0.095 ** |
| (z) | | | (2.48) |
| FisTrans | 0.002 *** | 0.002 *** | 0.002 *** |
| (z) | (3.96) | (3.94) | (3.72) |
| Market | 0.043 *** | 0.042 *** | 0.041 *** |
| (z) | (7.00) | (6.85) | (6.74) |
| PopDens | −0.486 *** | −0.465 *** | −0.467 *** |
| (z) | (−4.15) | (−3.98) | (−4.03) |
| HR | −0.168 ** | −0.161 ** | −0.173 ** |
| (z) | (−2.24) | (−2.13) | (−2.32) |
| FD | 0.273 *** | 0.283 *** | 0.288 *** |
| (z) | (4.81) | (4.97) | (5.10) |
| PopStr | 0.002 | 0.002 | 0.002 |
| (z) | (0.61) | (0.68) | (0.89) |
| IndStr | −0.894 *** | −0.894 *** | −0.882 *** |
| (z) | (−9.80) | (−9.53) | (−9.67) |
| Constant | 0.760 *** | 0.761 *** | 0.755 *** |
| (z) | (7.39) | (6.96) | (7.52) |
| Lambda | 0.017 *** | 0.017 *** | 0.015 *** |
| (z) | (3.60) | (3.53) | (3.23) |
| Sigma | 0.102 *** | 0.102 *** | 0.101 *** |
| (z) | (26.83) | (26.83) | (26.83) |
| LM-Lag | 0.704 | 0.729 | 0.745 |
| (*p*-Value) | (0.402) | (0.393) | (0.388) |
| LM-Err | 38.647 *** | 38.344 *** | 33.167 *** |
| (*p*-Value) | (0.000) | (0.000) | (0.000) |
| LR Test | 12.941 *** | 12.441 *** | 10.405 *** |
| (*p*-Value) | (0.000) | (0.000) | (0.001) |
| Adjusted-$R^2$ | 0.452 | 0.450 | 0.466 |
| Observations | 360 | 360 | 360 |

Note: ** significant at 5% level, and *** significant at 1% level.

*5.6. Coupling Effect Test*

Local fiscal risk is to some extent the result of low efficiency in fiscal expenditure accumulation. Improving local fiscal expenditure efficiency reduces risk. We conclude that effective national auditing improves local fiscal expenditure efficiency and can mitigate local fiscal risks in China. We took the local government fiscal deficit ratio as a proxy variable for local financial risk (FinRisk) to test how it is affected by the national audit as shown in Table 13. The regression coefficients of the three functions of the national audit and local government FinRisk are significantly negative, indicating that the national audit effectively reduces local financial risks.

**Table 13.** Analysis of economic consequences.

| Variables | (1) | (2) | (3) |
|---|---|---|---|
| | FinRisk | FinRisk | FinRisk |
| L.NauditDis | −0.106 *** | | |
| (z) | (−2.64) | | |
| L.NauditRes | | −0.069 *** | |
| (z) | | (−2.90) | |
| L.NauditPre | | | −0.121 *** |
| (z) | | | (−3.94) |
| FisTrans | −0.0003 | 0.0003 | −0.0002 |
| (z) | (−0.21) | (0.18) | (−0.16) |
| Market | −0.044 * | −0.019 | −0.022 |
| (z) | (−1.74) | (−0.70) | (−0.84) |
| PopDens | 0.975 * | 0.788 | 0.915 * |
| (z) | (1.92) | (1.53) | (1.82) |
| HR | −1.672 *** | −1.762 *** | −1.932 *** |
| (z) | (−5.34) | (−5.61) | (−6.09) |
| FD | −4.214 *** | −4.378 *** | −4.272 *** |
| (z) | (−17.39) | (−17.79) | (−17.76) |
| PopStr | 0.00003 | −0.002 | −0.003 |
| (z) | (0.00) | (−0.20) | (−0.32) |
| IndStr | 2.965 *** | 2.812 *** | 2.649 *** |
| (z) | (7.88) | (7.33) | (6.87) |
| Constant | 4.892 *** | 5.417 *** | 6.384 *** |
| (z) | (12.02) | (13.42) | (12.76) |
| Lambda | −0.013 *** | −0.010 *** | −0.006 ** |
| (z) | (−4.16) | (−3.39) | (−2.46) |
| Sigma | 0.442 *** | 0.441 *** | 0.437 *** |
| (z) | (26.83) | (26.83) | (26.83) |
| LM-Lag | 0.556 | 0.827 | 0.116 |
| (*p*-Value) | (0.456) | (0.363) | (0.733) |
| LM-Err | 32.442 *** | 41.699 *** | 31.289 *** |
| (*p*-Value) | (0.000) | (0.000) | (0.000) |
| LR Test | 17.297 *** | 11.471 *** | 6.071 ** |
| (*p*-Value) | (0.000) | (0.000) | (0.014) |
| Adjusted-$R^2$ | 0.772 | 0.776 | 0.784 |
| Observations | 360 | 360 | 360 |

Note: * significant at 10% level, ** significant at 5% level, and *** significant at 1% level.

## 6. Discussion

This paper examines the direct and intermediate effects of national audit on improving the efficiency of local fiscal expenditure.

Zheng Xiaorong and Zhang Lu verified the information transmission function of national audit, which aroused the supervision consciousness of other supervisory subjects in the society [32] and improved the media's attention to local fiscal expenditure. However, whether this information transmission function can further exert governance effect and improve the efficiency of local fiscal expenditure has not been tested. On this basis, by examining the mediating effect of media attention in the process of national audit to improve the efficiency of local fiscal expenditure, this paper verifies the governance effect of national audit information transmission function on local fiscal expenditure.

Guo Mengnan and Wu Qiusheng discovered the synergic governance effect of state audit and other supervisory bodies [49], but not much literature has discussed the coupling relationship between state audit and other supervisory bodies. This paper found that the coupling of state audit and media attention also had a positive effect on improving the efficiency of local fiscal expenditure, which was a supplement to previous studies.

This paper uses the spatial econometrics model to control the spatial dependence of local fiscal expenditure efficiency, which is a deepening of the existing research. The spatial correlation and spatial dependence exist in the efficiency of local fiscal expenditure. The spatial factors are included in the model, and the influence of spatial disturbance is eliminated, which makes the estimation of the model more accurate.

## 7. Conclusions and Policy Implications

Based on the panel data for 30 provinces in mainland China (not including the Tibet Autonomous Region) from 2007 to 2018, we used a spatial econometric model to examine the direct and mediating effects of national auditing in improving local fiscal expenditure efficiency. We found that the national audit, from within the state's larger supervisory system, can reveal, resist, and prevent the improper use of funds. The national audit reveals irregularities in spending and places pressure on auditees to utilize funds in an effective, safe, and sound manner. Media attention plays a mediating effect; the information transmission function of national auditing exerts governance. Coupling of the national audit and media attention also improves local fiscal expenditure efficiency. Our observations of economic consequences show that national auditing not only improves the efficiency of local fiscal expenditure but also reduces local fiscal risks. The results of this work may provide theoretical and empirical evidence for national auditors seeking to mitigate risk, promote sustainable financial development, and modernize national governance systems.

In addition to verifying the direct effect of the disclosing, resisting, and preventing functions of the national audit on the improvement of local fiscal expenditure efficiency, this study also verifies the mediating effect of media attention, indicating that the national audit not only plays the function of professional supervision but also plays the function of information transmission, promotes the "organic integration and coordination of all kinds of supervision", and further enhances the governance effect. It provides theoretical and empirical evidence for a national audit to defuse fiscal risks, promote sustainable fiscal development, and promote the modernization of national governance system and governance capacity.

We suggest that strengthening the functions of the national audit to disclose, resist, and prevent improper spending will improve local fiscal expenditure efficiency. By supervising all aspects of finance, we should innovate the manner in which compliance and fiscal audits are implemented, promote performance auditing, improve the accountability mechanism, protect legal compliance across fiscal activities, promote the rational allocation of fiscal resources, and ensure the standardized use of fiscal funds so as to achieve efficiency goals.

We further suggest strengthening the openness of audit information, optimizing the content and format of national audit announcements, increasing the disclosure of violations, and reducing the threshold under which the public understands and uses audit announcement information in a manner that generates public enthusiasm. Reforming the audit management system may allow it to actively adapt to the novel requirements of the modern national government.

Our article also has some limitations. First of all, the measurement method of local fiscal expenditure efficiency needs to be improved. Referring to the research of most scholars, this paper measures the efficiency of local fiscal expenditure by data envelopment analysis. However, this method requires high data quality, and the measurement results are easily affected by outliers. Therefore, other latest efficiency measurement methods can be adopted in future studies to improve the accuracy of efficiency measurement results. Secondly, this paper verifies the governance effect of national audit information transmission function in the financial field from the perspective of national audits arousing

media attention and improving the efficiency of local fiscal expenditure. Whether the national audit can stimulate the supervision enthusiasm of the public, media, social groups, and other social supervisory subjects through the function of information transmission and then produce the effectiveness of other national governance needs to be further studied. Finally, the coupling-effect test results of this paper show that the coupling of national audit and media attention also has a positive effect on improving the efficiency of local fiscal expenditure, indicating that the national audit can have a benign resonance with social supervision subjects and produce governance efficiency. However, this paper only tests the benign resonance between state audit and media attention. Whether the state audit has a benign resonance with other supervision needs to be further tested.

**Author Contributions:** Conceptualization, D.Z., X.S. and C.P.; methodology, D.Z., X.S. and C.P.; software, D.Z., X.S. and C.P.; validation, D.Z., X.S. and C.P.; formal analysis, D.Z. and X.S.; investigation, X.S.; resources, D.Z., X.S. and C.P.; data curation, D.Z. and X.S.; writing—original draft preparation, X.S.; writing—review and editing, D.Z., X.S. and C.P.; visualization, D.Z., X.S. and C.P.; supervision, D.Z.; project administration, D.Z.; funding acquisition, D.Z. All authors have read and agreed to the published version of the manuscript.

**Funding:** This research was supported by the National Social Science Foundation of China (grant number: 18BJY023).

**Institutional Review Board Statement:** Not applicable.

**Informed Consent Statement:** Not applicable.

**Data Availability Statement:** The data on the amount of funds found in violation by the state audit, the amount of funds handled by the state audit, and the amount of audit suggestions are from the China Audit Yearbook. The input data used to measure the efficiency of local fiscal expenditure are from China Fiscal Yearbook, and the output data are from the official website of National Bureau of Statistics. Media attention data refer to the number of media reports retrieved from China National Knowledge Network (CNKI) *Full-text Database of China's Major Newspapers* by using the names of provinces, autonomous regions, and municipalities directly under the central government and "finance" as keywords in the corresponding years. Fiscal transparency data come from the annual China Fiscal Transparency Report issued by Shanghai University of Finance and Economics. The institutional environment data come from Wang Xiaolu et al.'s book *China's Marketization Index by Province* (2021); other data came from the China Statistical Yearbook or the official website of the National Bureau of Statistics.

**Conflicts of Interest:** The authors declare no conflict of interest.

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
