# Peer review of "National Audit, Media Attention, and Efficiency of Local Fiscal Expenditure: A Spatial Econometric Analysis Based on Provincial Panel Data in China"

_sustainability, doi:10.3390/su15010532_

Round 1

Reviewer 1 Report

Compliments to the authors for the effort they have made to write the paper.

I think that the topic of this paper is timely and needed, but It is necessary to better connect the results with the previous studies on this topic.

The abstract adequately explain the motivation of the study and the main features of methodology, then summarize the findings.

Introduction and literature review. You need to better link the goal of your study to previous findings. You also need to provide more details about the recent developments of the research in this field.

Methodology. The hypothesis, sample selection and data sources are well presented, but the research methodology shoudl be improved by reorganyzing the section 3.Theoretical analysis and research hypothesis and 4. Research design. 

Results. The section entitled Empirical analysis highlights the results of the study, but I think the findings should be presented in relation to those of other studies.

Conclusions. Authors should show to what extent their results confirm or refute the results of other studies. Otherwise, the paper's findings seem too obvious and do not suggest a specific contribution to the literature.

 References. The bibliographic references must be more recent and representative.

Reviewer 2 Report

Zhang et al. applied Tobit spatial error model to test the impact of national auditing on local fiscal expenditure efficiency and to investigate the intermediary role of media attention. The article is properly designed and structured. Results are clearly presented, and conclusions are well supported. However, the theoretical contribution is insignificant. This article can be improved if some of the following issues are addressed.

·         Improper formatting. One space after punctuation. Line 28, 36, etc. Use English quotation marks. Line 25, 44, 49, etc.

·         Using past tense when you explain your analysis in the past. Line 54 – 57, etc.

·         It is odd to explain the contributions of the research in the introduction section, Line 61 – 67.

·         It is better to add a discussion section to summarize and further analyze empirical results presented in Section 5.

·         The article is hard to read. Extensive editing and proofreading are required. 

Reviewer 3 Report

The article "National Audit, Media Attention, and Efficiency of Local Fiscal Expenditure: A Spatial Econometric Analysis Based on Provincial Panel Data in China" addresses an interesting and current topic. 

The article is methodologically well conceived and theoretically supported.

However, we believe that the methodology should include the population and the sample analysed, since the population was not quantified and therefore we do not know if the sample is representative. This situation may limit the study.

The results of the study and respective discussion could be improved with greater depth of the results obtained and comparison with previous studies. We believe that here the authors can improve the work in order to support the results obtained.

Also, the conclusions were not supported by the theoretical framework mentioned above or by previous studies. The authors here should also corroborate, or not, their conclusions with previous studies.

Reviewer 4 Report

Dear Authors,

Let me congratulate you on publishing such an interesting article.

I would suggest you get the article reviewed by a native English speaker or teacher of English language.

There are a few issues I would like you to address:

1. Your hypothesis H1b is not very clear: What do you mean by National auditing resisting? How can resistance to auditing increase efficiency? Please use more appropriate words. You may consider changing the language in hypothesis H1c also.

2. In line 290, you mention telecommunications in the table. Does it include internet too? You combined posts and telecommunications. You might consider separating both or defining clearly what it means.

3. Similarly in the same table you put Harmless treatment rate of household garbage and social security under the same line. Are both combined or did you forget to separate them?

4. While your conclusions are sound and make sense, I would suggest you add one final paragraph on the limitations of your study and give some recommendations for further research, so that some other researchers may expand your work.
